# Self-Evaporation Dynamics of Quantum Droplets in a $^{41}$K-$^{87}$Rb Mixture

Chiara Fort [1,2] and Michele Modugno [3,4,*]

1    LENS and Dipartimento di Fisica e Astronomia, Università di Firenze, 50019 Sesto Fiorentino, Italy; chiara.fort@unifi.it

2    Istituto Nazionale di Ottica, CNR-INO, 50019 Sesto Fiorentino, Italy

3    Department of Physics, University of the Basque Country UPV/EHU, 48080 Bilbao, Spain

4    IKERBASQUE, Basque Foundation for Science, 48009 Bilbao, Spain

*    Correspondence: michele.modugno@ehu.eus

**Abstract:** We theoretically investigate the self-evaporation dynamics of quantum droplets in a $^{41}$K-$^{87}$Rb mixture, in free-space. The dynamical formation of the droplet and the effects related to the presence of three-body losses are analyzed by means of numerical simulations. We identify a regime of parameters allowing for the observation of the droplet self-evaporation in a feasible experimental setup.

**Keywords:** atomic mixtures; quantum droplets; Gross-Pitaevskii simulation





## 1. Introduction

As demonstrated in a seminal paper by D. Petrov [1], ultracold quantum gases can exist in the form of self-bound droplets that do not expand even in the absence of any confinement. This liquid-like behavior, which originates from the interplay of attractive mean-field interactions and the repulsive effect of quantum fluctuations [2–4], was successfully observed in a number of experiments with dipolar condensates [5–9], homonuclear mixtures of $^{39}$K [10–13], and recently in a heteronuclear mixture of $^{41}$K and $^{87}$Rb [14,15]. These findings have triggered an intense research activity on droplets properties (see, e.g., the recent reviews in Refs. [16,17]) and on the so-called Lee-Huang-Yang fluid [18–20], and have motivated further studies in low-dimensional systems [21,22] and also beyond Petrov's theory [23–27].

Besides being self bound, one of the most peculiar characteristics of these quantum objects is the fact that they can be self-evaporating. Namely, in certain regimes the droplet cannot sustain any collective mode, such that any initial excitation is completely dissipated (that is, evaporated) until the system relaxes into a droplet with a lower number of atoms. This remarkable feature, originally predicted in Ref. [1] for bosonic mixtures, has been so far elusive to the experimental detection. Indeed, the first pioneering experiments performed with bosonic mixtures [10,11] have shown that the homonuclear mixtures of $^{39}$K suffer from strong three-body losses that continuously drive the system out of equilibrium, and eventually lead to the depletion of the droplet. This behavior was then confirmed by the theoretical analysis reported in Ref. [28], where the complex dynamics taking place during the droplet formation is thoroughly described. There, it was shown that the evolution of the system is indeed dominated by the presence of three-body losses and by a continuous release of atoms to restore the proper population ratio, whereas the self-evaporation mechanism plays only a negligible role, if any.

In this respect, the recent experiment in Ref. [14] represents a promising setup in which the self-evaporation mechanism could be properly investigated. As a matter of fact, for a $^{41}$K-$^{87}$Rb mixture the regime of parameters for which self-bound droplets form is such that three-body losses are expected to be significantly suppressed. Indeed, in such

a mixture, droplets form at lower densities $n \sim (\delta g/g)^2 a^{-3}$, and this allows for a larger ratio $\tau_{\text{life}}/\tau \sim n^{-1}$, with $\tau_{\text{life}}$ and $\tau$ being respectively the lifetime (limited by three-body losses) and the characteristic time scale of the droplet dynamics [14]. Thus, the fact that the $^{41}$K-$^{87}$Rb mixture is characterized by larger scattering lengths with respect to the $^{39}$K mixture employed in Refs. [10,11] allows for lower densities and, therefore, longer lifetimes. Actually, in that experiment no appreciable effect of three-body losses was observed on the timescale of several tens of milliseconds.

Motivated by the previous discussion, in this paper we analyze the self-evaporation mechanism and the role of three-body losses on the collective excitations of a $^{41}$K-$^{87}$Rb quantum droplet. For the sake of conceptual clarity, and in view of the fact that quantum droplets at equilibrium are spherically symmetric objects, we shall restrict the analysis to the case of a spherically symmetric system. Although this assumption limits the study to the monopole collective mode, it is however sufficient to draw the general behavior of the system and to discuss the role of three-body losses.

The paper is organized as follows. In Section 2 we review the general formalism for describing quantum droplets in heteronuclear bosonic mixtures. Then, in Section 3 we discuss the preparation of the initial state, namely a droplet compressed by a harmonic trapping. The dynamics of the droplet after the release of the trap is then studied in Section 4 in the unitary case (no losses), and in the presence of three-body losses. Finally, in Section 5 we draw the conclusions.

## 2. Self-Bound Droplets

We consider a binary condensate of $^{41}$K and $^{87}$Rb atoms in the $|F = 1, m_F = 1\rangle$ state, as considered in the recent experiment in Ref. [14]. The two components will be indicated as 1 and 2, respectively. This system is described by the following Gross-Pitaevskii (GP) energy functional, including both the mean field term and the Lee-Huang-Yang (LHY) correction accounting for quantum fluctuations in the local density approximation: [29]

$$E = \sum_{i=1}^{2} \int \left[ \frac{\hbar^2}{2m_i} |\nabla \psi_i(\mathbf{r})|^2 + V_i(\mathbf{r}) n_i(\mathbf{r}) \right] d\mathbf{r} + \frac{1}{2} \sum_{i,j=1}^{2} g_{ij} \int n_i(\mathbf{r}) n_j(\mathbf{r}) d\mathbf{r} + \int \mathcal{E}_{\text{LHY}}(n_1(\mathbf{r}), n_2(\mathbf{r})) d\mathbf{r}, \quad (1)$$

where $m_i$ are the atomic masses, $V_i(\mathbf{r})$ the external potentials, and $n_i(\mathbf{r}) = |\psi_i(\mathbf{r})|^2$ the densities of the two components ($i = 1, 2$). The LHY correction reads [1]

$$\mathcal{E}_{\text{LHY}} = \frac{8}{15\pi^2} \left( \frac{m_1}{\hbar^2} \right)^{3/2} (g_{11} n_1)^{5/2} f \left( \frac{m_2}{m_1}, \frac{g_{12}^2}{g_{11} g_{22}}, \frac{g_{22} n_2}{g_{11} n_1} \right)$$

$$\equiv \kappa (g_{11} n_1)^{5/2} f(z, u, x), \quad (2)$$

with $\kappa = 8m_1^{3/2}/(15\pi^2\hbar^3)$ and $f(z, u, x) > 0$ being a dimensionless function of the parameters $z \equiv m_2/m_1$, $u \equiv g_{12}^2/(g_{11}g_{22})$, and $x \equiv g_{22}n_2/(g_{11}n_1)$ [1,29]. The mixture is completely characterized in terms of the intraspecies $g_{ii} = 4\pi\hbar^2 a_i/m_i$ ($i = 1, 2$), and interspecies $g_{12} = 2\pi\hbar^2 a_{12}/m_{12}$ coupling constants, where $m_{12} = m_1 m_2/(m_1 + m_2)$ is the reduced mass. The values of the homonuclear scattering lengths are fixed to $a_{11} = 62a_0$ (A. Simoni, private communication), $a_{22} = 100.4a_0$, whereas the heteronuclear scattering length $a_{12}$ is considered here as a free parameter, that can be tuned by means of Feshbach resonances [14]. The onset of the MF collapse regime corresponds to $\delta g = g_{12} + \sqrt{g_1 g_2} = 0$, at $a_{12}^c = -73.6a_0$ [30].

In free space ($V_i \equiv 0$), the equilibrium density of a droplet is obtained by requiring the vanishing of the total pressure [31], which yields [1]

$$n_1^0 = \frac{25\pi}{1024} \frac{1}{a_{11}^3} \frac{\delta g^2}{g_{11}g_{22}} f^{-2}\left(\frac{m_2}{m_1}, 1, \sqrt{\frac{g_{22}}{g_{11}}}\right) \tag{3}$$

$$n_2^0 = n_1^0 \sqrt{\frac{g_{11}}{g_{22}}}. \tag{4}$$

Following [1,29], we consider this function at the mean-field collapse $u = 1$, $f(z, 1, x)$. We note that the actual expression for $f$ can be fitted very accurately with the same functional form of the homonuclear case [19]

$$f\left(\frac{m_2}{m_1}, 1, \sqrt{\frac{g_{22}}{g_{11}}}\right) \simeq \left[1 + \left(\frac{m_2}{m_1}\right)^{3/5} \sqrt{\frac{g_{22}}{g_{11}}}\right]^{5/2}. \tag{5}$$

For a finite number of atoms the droplet has a finite size, and it can be effectively described by a single wave function that satisfies the following dimensionless equation [1]

$$\left[-\frac{1}{2}\nabla_{\tilde{r}}^2 - 3\widetilde{N}|\phi_0|^2 + \frac{5}{2}\widetilde{N}^{3/2}|\phi_0|^3\right]\phi_0 = \tilde{\mu}\phi_0, \tag{6}$$

with $\int |\phi_0|^2 d\tilde{r} = 1$. Remarkably, the above equation is characterized by a single dimensionless parameter $\widetilde{N}$ defined as

$$\widetilde{N} = N_i / \left(n_i^{(0)}\xi^3\right), \tag{7}$$

where $\sum_i N_i = N$ and $N_1/N_2 = \sqrt{g_{22}/g_{11}}$ [see Equation (4)]. Here $\xi$ represents the length scale of the droplet [1,25]

$$\xi = \hbar\left[\frac{3}{2}\frac{\sqrt{g_{11}}/m_2 + \sqrt{g_{22}}/m_1}{|\delta g|\sqrt{g_{11}}n_1^{(0)}}\right]^{1/2}, \tag{8}$$

and $n_i^{(0)}$ the equilibrium density of each component in the uniform case. We also remind that the wave functions of the two condensates forming the droplet are given by $\psi_i = \sqrt{n_i^{(0)}}\phi_0$.

## 3. Preparation of the Initial State

Following Ref. [14], we initially prepare the atomic mixture in the ground state of an optical dipole trap. In order to simplify the discussion, and motivated by the fact that quantum droplets at equilibrium are spherically symmetric objects, we assume (i) that the two condensates are initially prepared in a spherically symmetric potential $U_j^d(r)$ ($j = 1, 2$), (ii) that the differential vertical gravitational sag (due to the different masses of the two atomic species) can be exactly compensated, and (iii) that equilibrium droplets can be smoothly prepared without any appreciable effect of three body losses [14]. With these assumptions, which will be maintained throughout this work for easiness of calculations and conceptual clarity, both the ground state and the dynamical evolution can be obtained by solving spherically symmetric equations. We remark that this approach restricts the analysis to the excitations of the monopole mode only. Indeed, the excitation of surface modes (with angular momentum $\ell \neq 0$), which arise when the condensate is prepared in an asymmetric trap, is ruled out by the assumption of spherical symmetry. However, the presence of these modes is not expected to modify qualitatively the picture resulting from the following discussion.

The ground state of the system is obtained by minimizing the energy functional in Equation (1) by means of a steepest-descent algorithm [32]. Formally, this corresponds to

solve the following set of stationary Gross-Pitaevskii (GP) equations for two wave functions $\psi_j$ [33]

$$\begin{cases} \left[ -\dfrac{\hbar^2}{2m_1}\nabla_r^2 + U_1^d(r) + \mu_1(n_1, n_2) \right]\psi_1 = \mu_1\psi_1 \\[2ex] \left[ -\dfrac{\hbar^2}{2m_2}\nabla_r^2 + U_2^d(r) + \mu_2(n_1, n_2) \right]\psi_2 = \mu_2\psi_2, \end{cases} \tag{9}$$

where $\nabla_r^2$ represents the radial component of the Laplacian:

$$\nabla_r^2 = \frac{1}{r^2}\frac{\partial}{\partial r}\left(r^2\frac{\partial}{\partial r}\right) = \frac{\partial^2}{\partial r^2} + \frac{2}{r}\frac{\partial}{\partial r}. \tag{10}$$

The local chemical potentials $\mu_i(n_1, n_2)$ include both the usual mean-field term and the Lee-Huang-Yang (LHY) correction [1], namely

$$\mu_i(n_1, n_2) = g_{ii}n_i + g_{12}n_j + \frac{\delta E_{LHY}}{\delta n_i}, \quad i \neq j, \tag{11}$$

with $n_i(r) \equiv |\psi_i(r)|^2$. We recall that droplets form in the regime of parameters where the mean-filed term alone would lead to the collapse of the mixture, that is instead avoided thanks to the repulsive effect of quantum fluctuations.

As for the optical dipole potential experienced by the two atomic species, it can be written as $U_j^d(r) = U_{0j}I(r)$, where $U_{0j}$ is species-dependent ($j = 1, 2$) and it depends on the atomic species polarizability and the wavelength of the optical potential, and $I(r)$ represents the intensity of the laser beam. For the atomic mixture $^{41}$K-$^{87}$Rb and a wavelength $\lambda = 1064$ nm [14] one has $\alpha \equiv U_{02}/U_{01} \simeq 1.15$. Then, in the harmonic approximation, we have

$$U_j^d(r) = \frac{1}{2}m_j\omega_j^2 r^2, \tag{12}$$

with $\omega_1 = \omega_2\sqrt{m_2/(\alpha m_1)}$. The value of the trap frequency can be tuned in order to produce the desired amount of initial excitations. For low enough $\omega_j$, the droplet will be prepared close to the equilibrium configuration in free space, whereas a tight confinement obviously corresponds to the excitation of compressional modes. In this paper we shall use $\omega_2 = 2\pi \times 50$ Hz, which produces a not too lage excitation of the monopole mode (see later on).

The other free parameters of the system are the inter-species scattering length $a_{12}$, that we will consider in the range $a_{12} \in [-95, -73.6]a_0$, below to the mean-field collapse threshold, and the number of atoms of the two species. As for the latter, we shall vary $N_2$ in the range $[1.5, 15] \times 10^4$, keeping $N_1 = N_2\sqrt{g_{22}/g_{11}}$, so that the atom numbers match the nominal equilibrium ratio in Equation (4).

## 4. Dynamics

Once the initial state is prepared, the optical dipole trap is switched off and the system is let to evolve. The droplet dynamics is studied here by solving the following set of GP equations [13,28,34,35] (see Appendix A)

$$\begin{cases} i\hbar\partial_t\psi_1 = \left[ -\dfrac{\hbar^2\nabla_r^2}{2m_1} + \mu_1(n_1, n_2) - i\hbar\dfrac{K_3}{2}|\psi_2|^4 \right]\psi_1 \\[2ex] i\hbar\partial_t\psi_2 = \left[ -\dfrac{\hbar^2\nabla_r^2}{2m_2} + \mu_2(n_1, n_2) - i\hbar K_3|\psi_1|^2|\psi_2|^2 \right]\psi_2, \end{cases} \tag{13}$$

which include a dissipative term accounting for three-body losses in the dominant recombination channel K-Rb-Rb (with $K_3 = 7 \times 10^{-41}$ m$^6$/s, see Ref. [14]). The above equations can be obtained by means of a variational principle [33] from the energy functional corresponding to the one in Equation (1) plus the dissipative term $-(i/2)\hbar K_3 \int n_1(\mathbf{r}, t)n_2(\mathbf{r}, t)^2 d^3r$.

Since the droplet is prepared in a *compressed* configuration (owing to the presence of the trap), the droplet will start oscillating. According to discussion in Ref. [1], in the absence of three-body losses the dynamics is expected to be characterized either by sinusoidal oscillations of the droplet width, where the monopole mode exists, or by damped oscillations, in the so-called *self-evaporation* regime. The latter represents one of the remarkable properties of quantum droplets [1,28], and it takes place in a certain window of $\widetilde{N}$, where the excitation spectrum of the droplet lies entirely in the continuum. The nominal phase diagram for our system (obtained from the predictions Ref. [1]) is shown in Figure 1 as a function of the interspecies scattering length $a_{12}$ and of the (initial) number of atoms $N_2$. The specific combination of parameters used in the figure, $(\widetilde{N} - N_c)^{1/4}$, is introduced after Ref. [1]. The monopole mode is expected to be stable for $\widetilde{N} > 933.7$, and to evaporate for lower values of $\widetilde{N}$. In the window $94.2 < \widetilde{N} < 933.7$ other modes with $\ell \neq 0$ (not included in the present discussion) may appear, whereas below $\widetilde{N} < 94.2$ neither the monopole nor the surface modes ($\ell \neq 0$) can be sustained, such that the droplet is expected to evaporate any initial excitation [1]. Below $\widetilde{N}_c$, where a droplet cannot be formed, the mixture forms a so-called LHY fluid: in this regime the MF interactions almost cancel out ($\delta g / \sqrt{g_{11} g_{22}} \approx 0$), and the system is governed only by quantum fluctuations [18–20].

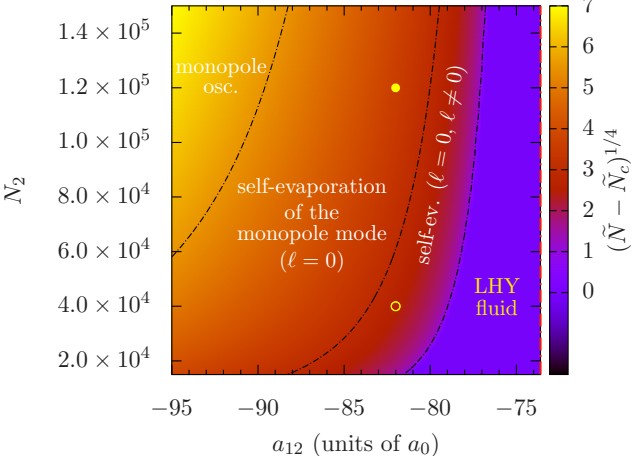

**Figure 1.** Heat map of $(\widetilde{N} - \widetilde{N}_c)^{1/4}$ as a function of $a_{12}$ and $N_2$, with $\widetilde{N}_c = 18.65$ being the critical value for the existence of a droplet [1]. The dotted-dashed lines, corresponding to $\widetilde{N} = 933.7$, 94.2, 18.65 (from left to right), represent the boundaries between the different regimes (see text). The vertical red dashed-dotted line at $a_{12} = -73.6$ corresponds to the onset of the MF collapse. The two circles at $a_{12} = -82 a_0$ refer to the parameter configurations considered in Section 4.

### 4.1. Estimate of the Droplet Lifetime

Before analyzing the detailed dynamical behavior of our system it is convenient to discuss the role of three-body losses on the droplet lifetime. To this aim, we shall consider a droplet in free space, at equilibrium (at $t = 0$). Then, *assuming* that the shape of the droplet is weakly affected by the atom losses (which is justified at the early stages at the evolution, at least), we can write

$$n_i(r, t) \simeq N_i(t) \rho(r), \tag{14}$$

where $N_1(0)/N_2(0) = \sqrt{g_{22}/g_{11}} \equiv \gamma$, and with $\rho(r)$ representing the droplet density profile normalized to unity [see Equations (4) and (7)]. With this in mind, the evolution of $N_i(t)$ can be obtained from the previous Equation (13) by neglecting the kinetic and chemical potential terms [which concur in determining the shape $\rho(r)$], left multiplying each equation by $\psi_i^*$, and integrating over the volume (see also Ref. [36]), yielding

$$\frac{d\bar{N}_1}{d\tau} = -\bar{N}_1 \bar{N}_2^2 \tag{15}$$

$$\frac{d\bar{N}_2}{d\tau} = -2\gamma\bar{N}_1\bar{N}_2^2, \tag{16}$$

where we have defined $\bar{N}_i(\tau) \equiv N_i(\tau)/N_i(0)$ and $\tau \equiv tK_3N_2^2(0)\int\rho^3d^3r$. Notice that the factor of 2 in Equation (16) corresponds to the fact that here we are considering the dominant recombination channel K-Rb-Rb (two atoms of Rb are lost for each atom of K).

The above equations have an approximate solution of the form

$$\bar{N}_1(\tau) = \frac{1-\beta}{1+(\tau/\tau_1)^{\alpha_1}} + \beta, \tag{17}$$

$$\bar{N}_2(\tau) = \frac{1}{1+(\tau/\tau_2)^{\alpha_2}}, \tag{18}$$

which is very accurate, indeed. In the present case ($\gamma \simeq 0.8736$), from a fit of the exact numerical solution of Equations (15) and (16) we find $\tau_1 \simeq 0.70$, $\alpha_1 \simeq 0.88$, $\beta \simeq 0.43$, $\tau_2 \simeq 0.71$, $\alpha_2 \simeq 0.86$. In particular, $\tau_2$ corresponds to the *half-life* of the $i = 2$ component (regardless of the value of $\alpha_2$), in which losses are dominant. Therefore, it represents a characteristic time through which we can measure the impact of three-body losses on the lifetime of the droplet. In our simple model the half-life is therefore $t_{1/2} \simeq \tau_2/[K_3N_2^2(0)\int\rho^3d^3r]$ that, besides the explicit dependence on $N_2(0)$, also depends implicitly on $a_{12}$ through the density distribution $\rho(r)$. The behavior of $t_{1/2}$ as a function of $a_{12}$ and $N_2(0)$ is shown in Figure 2. There we compare the prediction of the above analytical model with the actual values obtained from the solution of the GP equations in (13). The qualitative agreement is remarkable.

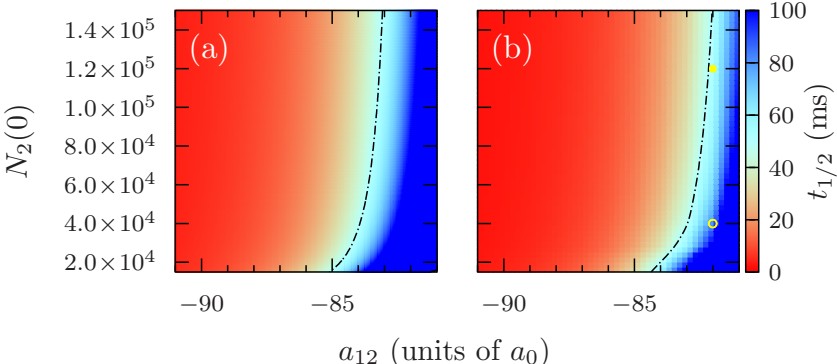

**Figure 2.** Heat map of $t_{1/2}$ as a function of $a_{12}$ and $N_2(0)$. (**a**) Prediction of the analytical model, see Equation (18); (**b**) values extracted from the actual decay of $N_2(t)$ obtained from the solution of the GP equations in (13). The color scale is saturated at $t_{1/2} = 100$ ms (dark blue). The dashed-dotted line corresponds to $t_{1/2} = 50$ ms. The two circles at $a_{12} = -82a_0$ refer to the parameter configurations considered in the GP simulations of Section 4.2.

### 4.2. Damped Monopole Oscillations

Given the above picture, in the following we shall investigate the self evaporation dynamics for $a_{12} = -82a_0$, where the droplet half-life is larger than 50 ms [see Figure 2b]. In particular, we consider two different configurations with $N_2 = 4 \times 10^4$ and $N_2 = 1.2 \times 10^5$, indicated by the yellow circles in Figures 1 and 2b, both lying in the self-evaporation regime. It is worth to remark that in the regime where the monopole mode is expected to be stable (see Figure 1) the droplet is affected by severe three-body losses (see Figure 2), which make the detection of this mode unfeasible.

In order to distinguish between the droplet density distribution and the distribution of atoms that evaporate, we define the *droplet volume* as that contained within a certain radius $R_d$. Then, the various droplet properties can be easily computed by integrating within the droplet volume. In the present case, this radius can be conveniently fixed to $R_d = 8$ μm, which is sufficiently large to include both the bulk and the droplet tails. The numerical simulations of the GP equations are performed on a computational box that is

at least one order of magnitude larger than $R_d$. Accordingly, with $N_i^d(t)$ we indicate the number of atoms of each species ($i = 1, 2$) within the droplet volume, at time $t$. Then, we define a *running* value of $\widehat{N}$ as [28]

$$\widetilde{N}_R(t) \equiv \frac{1}{\xi^3} \frac{N_1^d(t) + N_2^d(t)}{N_1 + N_2} = k \sum_{i=1}^{2} N_i^d(t),\qquad(19)$$

where $k \simeq 4.41 \times 10^{-3}$ for the current values of the scattering lengths.

The evolution of the system is shown in in Figure 3, where we plot the rms width $\sigma(t)$ of the droplet, the running value of $[\widetilde{N}_R(t) - N_c]^{1/4}$, the ratio $N_1^d(t)/N_2^d(t)$, and the fraction of evaporated atoms for each species $N_i^{ev}(t)/N_i$, with and without three-body losses. In both cases the mixture is initially prepared in the ground state of a dipole trap of frequency $\omega_2 = 2\pi \times 50$ Hz (we recall that $\omega_1 = \omega_2 \sqrt{m_2/(\alpha m_1)}$, see Section 3; in the present case $\omega_1 = 2\pi \times 67.9$ Hz).

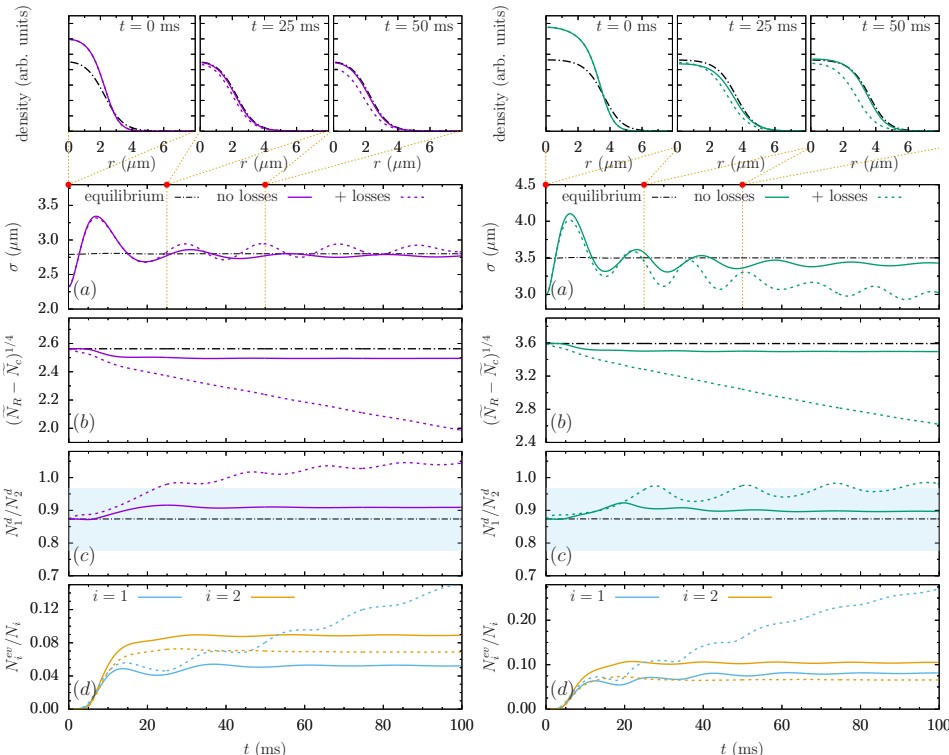

**Figure 3.** Evolution of (**a**) the droplet width $\sigma(t)$, (**b**) the running value of $\widetilde{N}_R(t)$ (see text), (**c**) the ratio $N_1^d(t)/N_2^d(t)$, and (**d**) the fraction of evaporated atoms for each species $N_i^{ev}(t)/N_i$, after the release from a trap of frequency $\omega_2 = 2\pi \times 50$ Hz, with and without three-body losses. The insets in the top row show the total density of the binary mixture, $n(r, t) = \sum_{i=1}^{2} N_i |\psi_i(r, t)|^2$, at different evolution times, corresponding to the red circles in (**a**). The horizontal line in (**b**) represent the *nominal* equilibrium value $N_1/N_2 = \sqrt{g_{22}/g_{11}} \simeq 0.873$, and the dashed area the corresponding tolerance (see text). Here $a_{12} = -82a_0$, $N_2 = 4 \times 10^4$ (left), $N_2 = 1.5 \times 10^5$ (right).

Let us first focus on the clean case, in the absence of three-body losses. As expected, we find that in both the investigated cases the droplet width performs damped sinusoidal oscillations (corresponding to a damped monopole mode), and the system eventually relaxes to an equilibrium configuration [see Figure 3a, along with the top panels], corresponding to a smaller, stationary value of $\widetilde{N}_R(t)$, see panels (b). This decrease in the number of particles is a consequence of the self-evaporation mechanism [1,28], which takes place within the first 15 ms. Looking at panel (d), where we plot the fraction of atoms of each species that are lost by self-evaporation, it is clear that the values of $N_i^{ev}(t)/N_i$ soon reach their asymptotic value (modulo small fluctuations). In addition, also the ratio between the atom numbers in

the two components remains close to the equilibrium value, see panels (c), with deviations below the reequilibration threshold (shaded area in the figure). Indeed, we recall that a droplet can sustain an excess of particles in one of the two components $\delta N_i^d / N_i^d$ up to a critical value $\sim \delta g / \sqrt{g_{11} g_{22}}$ [1] ($\approx 11\%$ in the present case), beyond which particles in excess are expelled.

Let us now discus how the presence of three-body losses affects the above picture. From panels (b) it is evident that losses produce a continuous drain of particles from the droplet. Nevertheless, after 100 ms of evolution the value of $\widetilde{N}_R(t)$ is still well above the critical value $\widetilde{N}_c$ for the existence of a droplet. Indeed, after several tens of milliseconds the mixture still forms a self-bound droplet (see e.g., the density profile at 50 ms) which keeps undergoing damped monopole oscillations, as shown in panels (a). Notably, the initial stage of the evolution is still dominated by self-evaporation, see panel (d), and then both the frequency and the amplitude of the oscillations become larger than in the clean case (without losses), because of the atoms that leave the bulk and populate the tails. Actually, there are two opposite effects taking place: on the one hand the droplet shrinks because of the loss of atoms due to three-body recombination, on the other hand there is an outward flow of K atoms that evaporate from the droplet, see panel (d). This species selective evaporation is due to the fact that the major loss of atoms affects the Rb component (see previous section), so that a progressively increasing fraction of K atoms cannot stay bound inside the droplet, and it is let free to expand. We remark that this is obviously a non-equilibrium process, as it is evident from the fact that $\widetilde{N}_R(t)$ does not relaxes to a stationary value and that the ratio $N_1^d(t) / N_2^d(t)$ soon run over the reequilibration threshold, anyway. All this is responsible for the different 'asymptotic' behavior of $\sigma(t)$ in the two cases shown in panels (a). For $N_2 = 4 \times 10^4$ (left), the two effects approximately compensate each other, and the width oscillates close to the nominal equilibrium value without losses. Instead, for $N_2 = 1.5 \times 10^5$ (right), which is characterized by a shorter lifetime, the droplet width decreases progressively (though keeping oscillating). It is worth to remark that the actual behavior is also sensitive to the choice of the droplet radius $R_d$.

## 5. Conclusions

We have theoretically investigated the self-evaporation dynamics of quantum droplets in a $^{41}$K-$^{87}$Rb mixture in feasible experimental setups, including the effects of three-body losses. The mixture is prepared in the ground state of a spherically symmetric harmonic trap, that is then released thus letting the system evolve in free space. The subsequent dynamics, characterized by the excitation of the monopole breathing mode, has been analyzed by solving the coupled Gross-Pitaevskii equations for the two components. For the estimated values of three-body losses ($K_3 = 7 \times 10^{-41}$ m$^6$/s for the dominant recombination channel K-Rb-Rb [14]), we find that by tuning the interspecies scattering length $a_{12}$, the lifetime of the system can be easily adjusted to be of the order of, or larger than, 100 ms. This makes $^{41}$K-$^{87}$Rb droplets much more robust than those realized with two hyperfine states of $^{39}$K, whose lifetime is limited to the order of 10 ms [10,11,28]. Such long lifetimes permits to follow the droplet dynamics for several tens of milliseconds, without any appreciable loss of resolution. In this scenario, we have found that the initial stage of the evolution is dominated by the self-evaporation mechanism even in the presence of three-body losses, and that the latter induce an interesting non equilibrium dynamics at later times. These findings make the experimental investigation of collective modes of self-bound droplets in $^{41}$K-$^{87}$Rb mixtures very promising.

**Author Contributions:** All authors have equally contributed. All authors have read and agreed to the published version of the manuscript.

**Funding:** This work was supported by the Spanish Ministry of Science, Innovation and Universities and the European Regional Development Fund FEDER through Grant No. PGC2018-101355-B-I00 (MCIU/AEI/FEDER, UE), by the Basque Government through Grant No. IT986-16.

**Institutional Review Board Statement:** Not applicable.

**Informed Consent Statement:** Not applicable.

**Acknowledgments:** We thank Alessia Burchianti, Luca Cavicchioli, and Francesco Minardi for the critical reading of the manuscript.

**Conflicts of Interest:** The funders had no role in the design of the study; in the collection, analyses, or interpretation of data; in the writing of the manuscript, or in the decision to publish the results.

## Appendix A. Numerical Methods

Let us consider a GP equation of the form

$$i\partial_\tau \varphi(r, \tau) = H\varphi(r, \tau), \tag{A1}$$

with

$$H = -\frac{1}{2}\nabla_r^2 + V(r) + g|\varphi(r, \tau)|^2, \tag{A2}$$

and

$$\nabla^2 = \frac{\partial^2}{\partial r^2} + \frac{2}{r}\frac{\partial}{\partial r}. \tag{A3}$$

The Crank-Nicholson algorithm (on a discrete space-time grid) consists in solving the following system of linear equations with the components of the vector $\varphi^{n+1} \equiv (\varphi_1, \ldots, \varphi_j, \ldots \varphi_N)^{n+1}$ being the unknown variables [32]

$$\left(1 + \frac{i\Delta\tau}{2}H\right)\varphi^{n+1} = \left(1 - \frac{i\Delta\tau}{2}H\right)\varphi^n. \tag{A4}$$

Notice that this algorithm preserves the unitarity for real-time evolutions.
Derivatives can be written in terms of central differences [37],

$$\frac{\partial^2\varphi}{\partial r^2} \to \frac{\varphi_{j+1} - 2\varphi_j + \varphi_{j-1}}{\Delta^2} \tag{A5}$$

$$\frac{2}{r}\frac{\partial\varphi}{\partial r} \to \frac{2}{r_j}\frac{\varphi_{j+1} - \varphi_{j-1}}{2\Delta} \tag{A6}$$

that allow to write the linear system Equation (A4) in the following tridiagonal form,

$$a_j\varphi_{j-1}^{n+1} + b_j\varphi_j^{n+1} + c_j\varphi_{j+1}^{n+1} = d_j. \tag{A7}$$

The explicit expression for the coefficients $a_j, b_j, c_j$, and $d_j$ $(j = 1, \cdots N)$ are

$$a_j = \frac{i\Delta\tau}{2}\left[\frac{1}{\Delta^2} - \frac{1}{r_j\Delta}\right], \tag{A8}$$

$$b_j = 1 + \frac{i\Delta\tau}{2}\left[-\frac{2}{\Delta^2} + V(r_j) + g|\varphi_j^n|^2\right], \tag{A9}$$

$$c_j = \frac{i\Delta\tau}{2}\left[\frac{1}{\Delta^2} + \frac{1}{r_j\Delta}\right], \tag{A10}$$

$$d_j = -a_j\varphi_{j-1}^n + (2 - b_j)\varphi_j^n - c_j\varphi_{j+1}^n, \tag{A11}$$

which have to be accompanied by suitable boundary conditions in order to preserve the tridiagonal form.

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
