# Peer review of "Self-Evaporation Dynamics of Quantum Droplets in a 41K-87Rb Mixture"

_applsci, doi:10.3390/app11020866_

Round 1
Reviewer 1 Report
This paper deals with the study of self-evaporation of quantum droplets in ultra-cold mixture of K and Rb, using numerical simulations. The aim was to identify regimes where self-evaporation would dominate over the usual three-body losses. The paper is interesting and well written. In my opinion it could be accepted for publication, after a few small amendments.
Here are my proposed changes:
- – The meaning of $\tilde N$ should be more clearly stated after eq. (6), given that it is essential to the description of the quantum droplet.
- – A short comment should be made on the relation between the static and the dynamic GP equations (9) and (13), noting their similarities and differences, given that they derive from the same theoretical model.
- – The dependence of $\tau_{1/2}$ with respect to $a_{12}$ should be made more explicit to the benefit of the reader. This would help to understand the meaning of fig.2.
- – It is difficult to understand why the droplet volume is defined by a fixed radius $R_d$, on page 7, and not by its actual size. How does it compare with the droplet width $\sigma ( t )$? The authors should clarify this point.
- – The inclusion of a new figure, containing images of the simulation itself, could help to explain the previous point. This is, of course, optional.
- – A short comment should be added on the possible mechanisms leading to the formation of quantum droplets, which are in some sense the reverse of the evaporation mechanisms discussed here.
Author Response
We thank the Referee for the careful reading of the manuscript and the useful comments and suggestions. Here we reply to all the points raised.
- The meaning of $\tilde N$ should be more clearly stated after eq. (6), given that it is essential to the description of the quantum droplet.
The quantity $\tilde N$ is defined by Eq. (7), which we have now written in a more explicit form. $\tilde N$ represents the only dimensionless parameter governing the properties of the droplet ground state. This has been now explicitly stated in the manuscript.
- A short comment should be made on the relation between the static and the dynamic GP equations (9) and (13), noting their similarities and differences, given that they derive from the same theoretical model.
We have now explicitly stated that both equations (9) and (13) are obtained from the energy functional in Eq. (1), with the inclusion of also a dissipative term in the latter case.
- The dependence of $\tau_{1/2}$ with respect to $a_{12}$ should be made more explicit to the benefit of the reader. This would help to understand the meaning of fig.2.
Unfortunately we have not an analytical expression for the dependence of $\tau_{1/2}$ on $a_{12}$. As we write in the text, the half-life depends implicitly on $a_{12}$ through the density distribution $\rho(r)$, and so it can be computed only numerically, as we do in Fig. 2.
- It is difficult to understand why the droplet volume is defined by a fixed radius $R_d$, on page 7, and not by its actual size. How does it compare with the droplet width $\sigma (t)$? The authors should clarify this point.
The droplet volume considered here and the one defined by the width $\sigma (t)$ correspond to two different concepts (see also the new reference [28]). The point is that the various quantities which characterize the droplet (its energy, the rms width $\sigma$, the number of atoms, etc.) have to be computed by integrating the local densities (of energy, particles, etc.) within a certain volume. That’s why we have to introduce a volume associated to the droplet in order to distinguish what belongs to the droplet and what is flowing outward (the products of the evaporation). The corresponding radius R has to be sufficiently larger than the droplet rms width in order to include the bulk and the droplet tails. In the revised manuscript we have clarified the definition of the droplet volume, in order to make this point clear.
- The inclusion of a new figure, containing images of the simulation itself, could help to explain the previous point. This is, of course, optional.
Some images of the density distribution at different evolution times are already shown in the top row of Fig. 3. These figures are very representative of the typical density configurations.
- A short comment should be added on the possible mechanisms leading to the formation of quantum droplets, which are in some sense the reverse of the evaporation mechanisms discussed here.
As highlighted in the very first lines of the Introduction, the general mechanism that leads to the formation of quantum droplets is “the interplay of attractive mean-field interactions and the repulsive effect of quantum fluctuations”. If the Referee instead refers to what happens during the dynamical preparation of a given experiment, then different effects may enter into play, depending on the specific experimental implementation, see e.g. the new Ref. 28 [Ferioli et al., PRR 2020]. For the sake of conceptual clarity, in the present paper we assume that a droplet can be created close to equilibrium, as experimentally demonstrated in Ref. 14 [D’Errico et al. PRR 2019]. In the revised manuscript we have included a couple of comments about this point, in Sect. 3.
Reviewer 2 Report
In this work, the authors perform the calculations of the droplet dynamics in a heteronuclear mixture relevant for ongoing experiments. The method is based on the simple extended Gross-Pitaevskii equation with the first order correction to the equation of state and an imaginary term accounting for losses, and should be expected to work sufficiently well for this kind of estimates. It is shown that this configuration is quite favourable and may allow for more elaborate experimental studies. I am curious about the three-body losses in the system. Does the rate coefficient depend strongly on the scattering lengths in this parameter range, or is it only a minor effect? Also, how would the results change if one takes into account that the loss takes place mainly from the most dense region of the gas? Finally, the authors write that the initial trap frequency can be tuned to produce different initial number of excitations, but it seems there is no discussion of the dependence of the dynamics on the initial compression, a comment would be in order here.
Author Response
We thank the Referee for the appreciation of our work and the interesting comments, which we address below.
I am curious about the three-body losses in the system.
Does the rate coefficient depend strongly on the scattering lengths in this parameter range, or is it only a minor effect?
This is an interesting point. In principle, one may expect that the value of K3 depends on the scattering length a12, indeed. However, we are not aware of any theoretical estimate or experimental measurement in the range considered in our work [see Wacker et al. PRL 117, 163201 (2016)], except for the value at a12=-77.8 a0 measured in [14] [D’Errico et al., PRR 1, 033155 (2019)]. In our manuscript the simulations shown in Fig. 3 are performed at a12=-82a0, a value that is pretty close to the former, and we do not expect dramatic changes there.
Also, how would the results change if one takes into account that the loss takes place mainly from the most dense region of the gas?
The dissipative term in the GP equation (13) depends on the density distribution of the mixture, so that they automatically take into account that the loss takes place mainly from the most dense region of the gas.
Finally, the authors write that the initial trap frequency can be tuned to produce different initial number of excitations, but it seems there is no discussion of the dependence of the dynamics on the initial compression, a comment would be in order here.
We thank the Referee for pointing out this inconsistency. Actually it was not our aim to discuss the dependence of the dynamics on the initial compression, that would require an additional analysis. Our aim here is to discuss how the self evaporation mechanism affects the monopole mode in a regime of not too large excitations. We have added the paragraph after Eq. (12) in order to avoid confusion.